published: 01 September 2021
# Mental Health and Wellbeing of 9–12-year-old Children in Northern Canada Before the COVID-19 Pandemic and After the First Lockdown

*Julia Dabravolskaj[1†], Mohammed K. A. Khan[2†], Paul J. Veugelers[1*†] and Katerina Maximova[3,2†]*

[1]*School of Public Health, University of Alberta, Edmonton, AB, Canada, [2]Dalla Lana School of Public Health, University of Toronto, Toronto, ON, Canada, [3]MAP Centre for Urban Health Solutions, Li Ka Shing Knowledge Institute, St. Michael's Hospital, Toronto, ON, Canada*

*Edited by:*
*Franco Mascayano,*
*Columbia University Irving Medical Center, United States*

*Reviewed by:*
*Anthony Jehn,*
*Western University, Canada*
*Sally McManus,*
*City University of London, United Kingdom*

*\*Correspondence:*
*Paul J. Veugelers*
*paul.veugelers@ualberta.ca*

*†ORCID:*
*Julia Dabravolskaj*
*orcid.org/0000-0002-4420-668X*
*Mohammed KA Khan*
*orcid.org/0000-0003-2354-971X*
*Paul J. Veugelers*
*orcid.org/0000-0001-8996-0822*
*Katerina Maximova*
*orcid.org/0000-0001-9842-1927*

*Citation:*
*Dabravolskaj J, Khan MKA, Veugelers PJ and Maximova K (2021) Mental Health and Wellbeing of 9–12-year-old Children in Northern Canada Before the COVID-19 Pandemic and After the First Lockdown. Int J Public Health 66:1604219. doi: 10.3389/ijph.2021.1604219*

**Objectives:** Children's mental health and wellbeing declined during the first COVID-19 lockdown (Spring 2020), particularly among those from disadvantaged settings. We compared mental health and wellbeing of school-aged children observed pre-pandemic in 2018 and after the first lockdown was lifted and schools reopened in Fall 2020.

**Methods:** In 2018, we surveyed 476 grade 4–6 students (9–12 years old) from 11 schools in socioeconomically disadvantaged communities in Northern Canada that participate in a school-based health promotion program targeting healthy lifestyle behaviours and mental wellbeing. In November-December 2020, we surveyed 467 grade 4–6 students in the same schools. The 12 questions in the mental health and wellbeing domain were grouped based on correlation and examined using multivariable logistic regression.

**Results:** There were no notable changes pre-pandemic vs. post-lockdown in responses to each of the 12 questions or any of the sub-groupings.

**Conclusion:** Supporting schools to implement health promotion programs may help mitigate the impact of the pandemic on children's mental health and wellbeing. The findings align with recent calls for schools to remain open as long as possible during the pandemic response.

Keywords: health promotion, public health, children, social determinansts of health, COVID–19, internalizing problems, school health, mental health and wellbeing

## INTRODUCTION

On March 11, 2020 the World Health Organization declared the COVID-19 pandemic. While implemented public health measures were critical to combat the viral spread, 1.6 billion children have been affected by school closures, cancellations or modification of organized sport and recreation activities, and the enforcement of physical distancing and stay-at-home orders [1]. Although school-aged children are at a considerably lower risk of severe COVID-19 illness [2],

**TABLE 1** | Characteristics of 9–12-year-old students before the pandemic and after the first lockdown and school reopening, Canada, 2018–2020.

| Student-level: n | 2018 | 2020 |
|---|---|---|
| | n = 476 | n = 467 |
| Sex: n (%) | | |
| Girls | 237 (50) | 257 (55) |
| Boys | 239 (50) | 210 (45) |
| Grade: n (%) | | |
| Grade 4 | 150 (32) | 131 (28) |
| Grade 5 | 157 (33) | 146 (31) |
| Grade 6 | 169 (36) | 190 (41) |
| MHW score: n (%) | | |
| −12 to 0 | 81 (17) | 87 (19) |
| >0 to 12 | 395 (83) | 380 (81) |
| Language(s) spoken at home | | |
| English only | 353 (74) | 326 (70) |
| English and Indigenous | 86 (18) | 75 (16) |
| English and other | 37 (8) | 66 (14) |
| **School-level: n** | **n = 11** | |
| Region of residence[a]: n (%) | | |
| Rural | 8 (73) | |
| Small population centre | 3 (27) | |
| Material deprivation quintile: n (%) | | |
| 1 | 0 (0) | |
| 2 | 3 (27) | |
| 3 | 3 (27) | |
| 4 | 1 (10) | |
| 5 | 4 (36) | |
| Social deprivation quintile: n (%) | | |
| 1 | 4 (36) | |
| 2 | 2 (18) | |
| 3 | 2 (18) | |
| 4 | 3 (27) | |
| 5 | 0 (0) | |

MHW, mental health and wellbeing.
[a]Rural refers to a community with <1,000 population and Small PC refers to a community with 1,000–29,999 population.[23].

early reports indicate that the pandemic–through associated public health measures–jeopardized children's mental health and wellbeing during the first lockdown in Spring 2020 [3]. The burden of mental illness among children was high even before the pandemic, with as many as 20% of youth living with a mental disorder in Canada [4]. The pandemic added myriad stressors that impact children's mental health and wellbeing (e.g., fear of contracting or having loved ones contract COVID-19, feelings of loneliness and hopelessness, and the inability to seek comfort from extended family members and friends) [5]. Among 385 Canadian children and adolescents, 67–70% experienced deterioration in at least one mental health domain during the first lockdown, particularly among those with pre-existing mental health problems [6], while the rates of mental disorders in 5–16 year old children in the United Kingdom increased from 11% in 2017 to 16% in July 2020 [7]. The risk for poor mental health and wellbeing is even higher for children from socioeconomically disadvantaged settings who may experience additional stressors such as household food insecurity, parental loss of jobs or income [8], and disruptive family dynamics [9], leaving children

even more vulnerable when schools are closed as part of the COVID-19 response.

Schools play a critical role in the social development of children and, under normal circumstances, schools are where children spend the majority of their waking hours, where families connect before and after school hours, and are an integral setting of social support in the community. For children from vulnerable settings, schools can also act as a "protective layer" through the delivery of school-based programming, including nutrition programs and mental health support [10]. In case of school closure, the daily routine and social interactions are severely disrupted, leaving children at a potentially higher risk of loneliness and mental health problems [11]. Emerging evidence reports on the increase in depression symptoms and psychological distress in general populations of school-aged children during the first lockdown in Spring 2020 [3, 5, 6]. An important and timely question is whether the mental health and wellbeing of school-aged children resemble the pre-pandemic levels when children return to school, particularly among children from socioeconomically disadvantaged areas. In 11 rural and remote Northern communities in Canada, we compared mental health and wellbeing of school-aged children observed pre-pandemic in 2018 and after the first lockdown was lifted and schools reopened in Fall 2020.

## METHODS

In the Spring of 2018, we surveyed 9–12 year old students in grade 4–6 in 15 schools in rural and remote Northern communities in Canada. All schools participate in APPLE Schools (A Project Promoting healthy Living for Everyone in schools)—an innovative, internationally recognized, not-for-profit health program that promotes healthy lifestyle behaviours and mental wellbeing in schools located in socioeconomically disadvantaged communities [12]. Grounded in a Comprehensive Schools Health approach [13], APPLE Schools program helps to transform the school's culture to "make the healthy choice the easy choice" [14–18]. From November 4 to December 9, 2020, we repeated this survey among grade 4–6 students in 11 of these 15 schools: one school declined participation and three other schools were closed due to COVID-19 outbreaks during the data collection. Eight schools are located in rural (<1,000 people) and three in small population centres (1,000–29,999 people) (**Table 1**). According to the 2016 Canadian Census data, population in these communities ranged from 414 to 14,961. Of the rural schools, two are located on an Indian reserve or settlement, and one community is "fly-in" only. Data were collected in school during regular class time. Students logged into the online survey portal on their Chromebooks, using a unique username and password that was assigned to them. Data collection shifted from the in-person mode in 2018 (trained research assistants travelled to schools to administer the surveys) to the online mode in accordance with the COVID-19 protocols (trained research assistants connected to each classroom through Zoom to prompt the survey questions that were projected on the whiteboard in the classroom). A total of 476 and 467 students

completed the survey, with the participation rates of 71.5 and 78% in 2018 and 2020, respectively. All students provided assent and their parents/guardians provided active-information passive-permission consent. The Health Research Ethics Board of the University of Alberta (Pro00061528) and participating school boards approved all the procedures.

As part of the pandemic response, all in-school learning was suspended for K-12 schools on March 16 in Northwest Territories, March 17 in Alberta, and March 23 in Manitoba. Although school buildings were closed, online learning continued until June 2020. Since all 11 schools fall under the jurisdiction of respective provincial and territorial governments, they re-opened in September 2020, with enhanced public health measures to prevent viral transmission in schools, including mandatory masking and physical distancing, small group cohorting, and curricula adaptations (e.g., physical education classes).

As part of a survey on lifestyle behaviours (e.g., physical activity, sedentary behaviour, sleep, healthy eating), students completed a series of 12 questions in the domain of mental health and wellbeing suitable for grade 4–6 students, derived from population survey instruments [19–22]. These included five negatively phrased items: I worry a lot; I feel unhappy or sad; I have trouble paying attention; I have trouble enjoying myself; and I am in trouble with my teacher(s). Seven positively phrased items were: My future looks good to me; I have someone I trust to go to for advice; I do well in my schoolwork; I feel like I have many friends; I feel like I belong at schools; I like myself; and I like the way I look. For ease of interpretation, we centred the distribution around zero: response options ("never or almost never," "sometimes," and "often or almost always") were assigned a score of "−1", "0" and "1" for positively stated items and reverse coded for negatively stated items. The items were summed to create a cumulative score, ranging from −12 to +12, with higher values indicating better mental health and wellbeing (henceforward referred to as MHW score). Finally, the score was dichotomized using "0" as the cut-off value, with values above 0 representing better mental health and wellbeing, and between −12 and 0 (inclusive) worse mental health and wellbeing.

Students self-reported their sex (girl vs. boy). Covariates included student's grade, region of residence, and quintiles of area-based social and material deprivation indices. Region of residence was defined according to Statistics Canada classification as rural (<1,000 people) or small population centre (1,000–29,999 people) [23]. Social and material deprivation indices were derived from 2016 Canada Census data based on postal codes, with higher quintiles indicating higher deprivation (detailed procedures are available elsewhere [24]). The polychoric correlation between the quintiles of these indices (−0.26) was estimated to rule out concerns about collinearity. To ensure sufficient number of students and schools in each category, quintiles were combined to create a binary variable indicating lower vs. higher deprivation (1–3 vs. 4-5 for material deprivation and 1–2 vs. 3-5 for social deprivation for material deprivation).

Differences in responses to each of the 12 items in 2018 and 2020 were examined using a chi-square test with a Bonferroni correction (p-values below 0.004 were considered statistically significant), separately for girls and boys. In all other analyses

p-values below 0.05 were considered statistically significant. A student's t-test was used to examine differences in the cumulative scores in 2018 and 2020. The distribution of the scores in 2018 and 2020 was depicted using histograms and Kernel density plots. A multivariable logistic regression model was applied to compare the odds of having worse vs. better mental health and wellbeing in 2020 compared to 2018 after adjustment for covariates.

To acknowledge that certain clusters of items may reveal temporal changes in mental health and wellbeing that are not necessarily captured in the MHW score, gender-stratified exploratory factor analysis with varimax (orthogonal) rotation was employed on the 2018 observations to extract latent factors that maximized the explained variance. Following examination of the scree plot and based on the Kaiser criterion (eigenvalue >1), three clusters were identified separately for girls and boys. Factor scores were then calculated using the Bartlett's test of sphericity and dichotomized with scores greater than 0 representing doing better and those 0 and below doing worse in each of these clusters. Logistic regression was applied to estimate the odds of doing worse in 2020 compared to 2018 for the sub-grouping identified in each of the three clusters. For the missing values, we performed multiple imputation using multivariate imputation by chain equations (MICE) [25]. Approximately 91% of students provided responses to 12 items, and approximately 97% completed at least ten of the items. Three and 16 students who did not respond to six or more items in 2018 and 2020, respectively, were excluded from analyses. To acknowledge the nested data structure (students within schools), mixed effects models with a school-level random effect were tested. However, since the intra-class correlation appeared low (ranging between 0 and 0.02 for all models), we applied fixed effects regression models rather than mixed effects models as per established recommendations [26]. Analyses were conducted in R 4.0.2 software (GNU General Public License).

## RESULTS

**Table 1** shows the characteristics of the participants and school communities before the pandemic (2018) and after the first lockdown was lifted (2020). While an equal number of girls and boys participated in 2018 survey, there were slightly more girls (56%) participating in the 2020 survey. Compared to the 2018 survey before the pandemic, fewer grade four (27 vs. 32%) and more grade six (41 vs. 36%) students participated in the survey during the pandemic. Four and three schools are located in the most materially and socially deprived areas, respectively.

**Table 2** depicts the student responses to the 12 items in 2018 and 2020. For none of the items, the differences between 2018 and 2020 were either substantial or statistically significant. Slightly more boys responded "often or almost always" to two items: "I feel like I belong in school" (53% in 2020 vs. 45% in 2018) and "I have trouble enjoying myself" (58% in 2020 vs. 49% in 2018). The differences in girls' responses were less pronounced.

Based on the MHW score, there was a modest decline in mental health and wellbeing among girls (4.9 [SD = 4.6] in 2018 vs. 4.4 [SD = 4.5] in 2020), however it did not attain statistical

**TABLE 2 |** Proportions[a] of 9–12-year-old student responses to mental health and wellbeing items before the pandemic and after the first lockdown and school reopening, separately for girls and boys, Canada, 2018–2020.

| | Girls | | | Boys | | |
|---|---|---|---|---|---|---|
| | 2018 *n* (%) | 2020 *n* (%) | *p*-value[b] | 2018 *n* (%) | 2020 *n* (%) | *p*-value[b] |
| Negatively stated items | | | | | | |
| I feel unhappy or sad | | | 0.25 | | | 0.50 |
| Often or almost always | 27 (11) | 38 (15) | | 21 (9) | 25 (12) | |
| Sometimes | 140 (59) | 158 (61) | | 133 (56) | 115 (55) | |
| Never | 70 (30) | 61 (24) | | 85 (36) | 68 (33) | |
| I worry a lot | | | 0.22 | | | 0.57 |
| Often or almost always | 58 (25) | 82 (32) | | 52 (22) | 39 (19) | |
| Sometimes | 101 (43) | 103 (40) | | 88 (37) | 72 (35) | |
| Never | 74 (32) | 70 (27) | | 98 (41) | 94 (46) | |
| I am in trouble with my teacher(s) | | | 0.36 | | | 0.15 |
| Often or almost always | 14 (6) | 9 (4) | | 21 (9) | 21 (10) | |
| Sometimes | 44 (19) | 43 (17) | | 98 (41) | 67 (33) | |
| Never | 179 (76) | 203 (80) | | 118 (50) | 118 (57) | |
| I have trouble paying attention | | | 0.87 | | | 0.29 |
| Often or almost always | 36 (15) | 42 (17) | | 40 (17) | 36 (17) | |
| Sometimes | 100 (43) | 111 (44) | | 119 (50) | 87 (42) | |
| Never | 98 (42) | 99 (39) | | 79 (33) | 83 (40) | |
| I have trouble enjoying myself | | | 0.20 | | | 0.15 |
| Often or almost always | 25 (11) | 41 (16) | | 36 (15) | 37 (18) | |
| Sometimes | 77 (33) | 84 (33) | | 63 (27) | 69 (33) | |
| Never | 132 (56) | 132 (51) | | 136 (58) | 101 (49) | |
| Positively stated items | | | | | | |
| My future looks good to me | | | 0.35 | | | 0.44 |
| Never | 10 (4) | 17 (7) | | 9 (4) | 13 (6) | |
| Sometimes | 93 (40) | 109 (43) | | 94 (40) | 85 (40) | |
| Often or almost always | 132 (56) | 129 (51) | | 134 (57) | 112 (53) | |
| I like the way I look | | | 0.98 | | | 0.72 |
| Never | 24 (10) | 25 (10) | | 21 (9) | 23 (11) | |
| Sometimes | 96 (41) | 106 (42) | | 74 (31) | 62 (30) | |
| Often or almost always | 116 (49) | 124 (49) | | 143 (60) | 125 (60) | |
| I like myself | | | 0.72 | | | 0.20 |
| Never | 19 (8) | 22 (9) | | 18 (8) | 8 (4) | |
| Sometimes | 64 (27) | 77 (30) | | 62 (26) | 62 (30) | |
| Often or almost always | 151 (65) | 155 (61) | | 155 (66) | 137 (66) | |
| I feel like I belong at school | | | 0.77 | | | 0.05 |
| Never | 28 (12) | 34 (13) | | 48 (20) | 25 (12) | |
| Sometimes | 90 (38) | 104 (40) | | 83 (35) | 72 (35) | |
| Often or almost always | 117 (50) | 119 (46) | | 107 (45) | 111 (53) | |
| I do well in my schoolwork | | | 0.11 | | | 0.07 |
| Never | 5 (2) | 15 (6) | | 11 (5) | 4 (2) | |
| Sometimes | 93 (39) | 97 (38) | | 95 (40) | 104 (50) | |
| Often or almost always | 139 (59) | 144 (56) | | 131 (55) | 102 (49) | |
| I feel like I have many friends | | | 0.88 | | | 0.86 |
| Never | 26 (11) | 33 (13) | | 15 (6) | 14 (7) | |
| Sometimes | 81 (35) | 86 (34) | | 69 (29) | 65 (31) | |
| Often or almost always | 127 (54) | 136 (53) | | 153 (65) | 129 (62) | |
| If I have problems, there is someone I trust to go to for advice | | | 0.65 | | | 0.74 |
| Never | 32 (14) | 42 (16) | | 36 (15) | 32 (16) | |
| Sometimes | 75 (32) | 76 (30) | | 83 (35) | 79 (39) | |
| Often or almost always | 129 (55) | 138 (54) | | 118 (50) | 94 (46) | |

[a]*Excludes missing values.*
[b]*p-values are from chi-squared tests for independence and adjusted for multiple comparisons using Bonferroni test.*

significance. There was no difference in the MHW score among boys: 4.8 (SD = 4.4) in 2018 vs. 4.8 (SD = 4.4) in 2020. The distribution of the scores (**Figure 1**) shows modest differences around the mean among girls and boys, with very little differences at the extremes of the distribution. After adjustment for potential confounders, the differences between 2018 and 2020 remained small and not statistically significant, for both girls (odds ratio

[OR] 1.05, 95% confidence interval [CI] 0.82–1.33) (**Table 3**) and boys (OR 1.01, 95% CI 0.79–1.29) (**Table 4**).

Three unique clusters of the 12 items were identified (**Supplementary Table S1**). The three factors among girls were labelled as self-perception, social perception, internalizing and functioning problems. The three factors among boys were labelled as social and self-perception, functioning problems, and

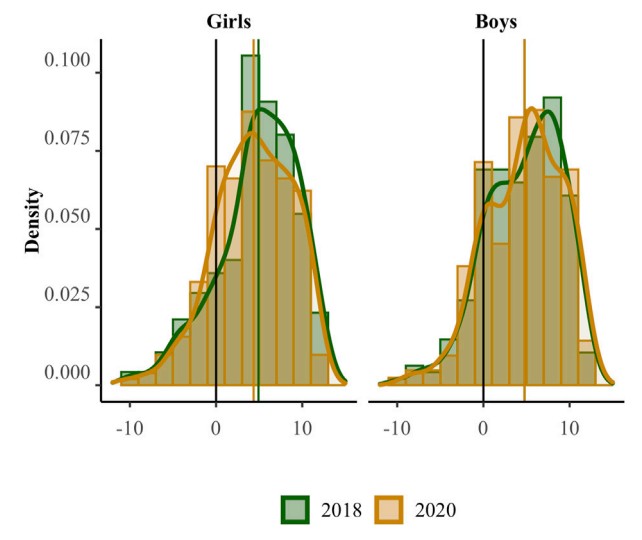

**FIGURE 1 |** Distribution* of the cumulative mental health and well being score before the pandemic and after the first lockdown and school reopening, separately for girls and boys, Canada, 2018–2020. *Based on Kernel density estimator. Changes in mean scores: −0.57 (95% CI -1.38, 0.23) in girls; 0.01 (95% CI -0.81, 0.81) in boys.

## DISCUSSION

The present study revealed the lack of notable pre-pandemic vs. post-lockdown differences in mental health and wellbeing in a specific segment of children residing in rural and remote, socioeconomically disadvantaged communities. Evidence is emerging on the impact of the pandemic on mental health and wellbeing in the general populations of children and youth *during* the first lockdown. A recent study in a convenience sample of 168 school-aged children aged 7.6–11.6 years in the UK found a significant increase in depression symptoms, but not anxiety symptoms or emotional problems, during the first phase of the UK lockdown (April-June 2020), compared to the initial assessment 18 months prior to the lockdown [3]. In another study that collected responses to an online survey during a similar timeframe (May-June 2020) in Germany, 1,040 children and adolescents between 11 and 17 years old self-reported significantly more mental health problems (17.8 vs. 9.9%) and higher anxiety levels (24.1 vs. 14.9%) than in another representative sample of German adolescents ($n = 1,556$) assessed well before the pandemic, in 2017. Compared to their counterparts from higher socioeconomic backgrounds, children and youth from low socioeconomic backgrounds, recent migrants or refugees, and those with constrained living arrangements were at a significantly higher risk of experiencing anxiety and depressive symptoms during the lockdown [27].

Our findings of no substantial impact on children's mental health and wellbeing post-lockdown are encouraging. We note that we surveyed students in November-December 2020—two to 3 months after schools were reopened. This window may have provided students enough time to adjust to the public health measures put in place to limit in-school transmission, while stressors related to school closure, being at home and separated

internalizing problems. There were no statistically significant differences between 2018 and 2020 in any of the sub-groupings identified (**Tables 3**, **4**). Finally, relative to those in grade 4, girls in grade 6 were more likely to report worse mental health and wellbeing and worse self-perception, whereas boys in grade 6 were more likely to report worse mental health and wellbeing and worse social and self-perception (**Table 3** and **4**).

**TABLE 3 |** Odds ratios[a] (95% CI) of worse mental health and wellbeing[b] after the first lockdown and school reopening compared to before the pandemic among 9–12 year old girls, Canada, 2018–2020.

| | MHW score | Self-perception | Social perception | Internalizing and functioning problems |
|---|---|---|---|---|
| | OR (95% CI) | OR (95% CI) | OR (95% CI) | OR (95% CI) |
| Year | | | | |
| 2018 | 1 | 1 | 1 | 1 |
| 2020 | 1.05 (0.82, 1.33) | 0.94 (0.78, 1.13) | 1.09 (0.91, 1.31) | 1.15 (0.96, 1.37) |
| Grade | | | | |
| Grade 4 | 1 | 1 | 1 | 1 |
| Grade 5 | 2.05 (0.96, 4.66) | 1.85 (1.12, 3.09) | 0.97 (0.60, 1.56) | 1.36 (0.85, 2.18) |
| Grade 6 | 3.99 (1.99, 8.75) | 2.61 (1.61, 4.29) | 0.68 (0.48, 1.08) | 1.33 (0.84, 2.11) |
| Region of residence | | | | |
| Rural | 1 | 1 | 1 | 1 |
| Small PC | 0.86 (0.39, 1.95) | 2.12 (1.14, 4.02) | 1.80 (0.99, 3.31) | 0.94 (0.52, 1.71) |
| Social deprivation | | | | |
| Lower | 1 | 1 | 1 | 1 |
| Higher | 1.31 (0.73, 2.42) | 0.72 (0.45, 1.15) | 0.76 (0.48, 1.20) | 0.93 (0.60, 1.47) |
| Material deprivation | | | | |
| Lower | 1 | 1 | 1 | 1 |
| Higher | 0.78 (0.30, 1.97) | 1.53 (0.77, 3.05) | 2.92 (1.50, 5.77) | 0.92 (0.48, 1.76) |

CI, confidence interval; MHW, mental health and wellbeing; OR, odds ratio; PC, population centre.
[a]Odds Ratios from multivariable logistic regression model and adjusted for grade level, social and material deprivation quintiles.
[b]Defined as score ranging from −12 to 0 (inclusive), or factor scores <0.

**TABLE 4 |** Odds ratios[a] (95% CI) of worse mental health and wellbeing[b] after the first lockdown and school reopening compared to before the pandemic among 9–12 year old boys, Canada, 2018–2020.

|  | MHW score | Social and self-perception | Functioning problems | Internalizing problems |
|---|---|---|---|---|
|  | OR (95% CI) | OR (95% CI) | OR (95% CI) | OR (95% CI) |
| Year |  |  |  |  |
| 2018 | 1 | 1 | 1 | 1 |
| 2020 | 1.01 (0.79, 1.29) | 0.97 (0.80, 1.17) | 0.89 (0.74, 1.08) | 1.11 (0.92, 1.33) |
| Grade |  |  |  |  |
| Grade 4 | 1 | 1 | 1 | 1 |
| Grade 5 | 1.99 (1.05, 3.85) | 1.06 (0.66, 1.72) | 1.17 (0.73, 1.86) | 0.69 (0.43, 1.09) |
| Grade 6 | 2.48 (1.32, 4.78) | 1.91 (1.19, 3.08) | 0.91 (0.57, 1.46) | 1.06 (0.66, 1.69) |
| Region of residence |  |  |  |  |
| Rural | 1 | 1 | 1 | 1 |
| Small PC | 0.88 (0.41, 1.94) | 1.06 (0.59, 1.93) | 0.88 (0.49, 1.56) | 0.79 (0.44, 1.41) |
| Social deprivation |  |  |  |  |
| Lower | 1 | 1 | 1 | 1 |
| Higher | 1.19 (0.67, 2.16) | 0.89 (0.56, 1.42) | 1.07 (0.68, 1.70) | 1.26 (0.80, 2.00) |
| Material deprivation |  |  |  |  |
| Lower | 1 | 1 | 1 | 1 |
| Higher | 1.28 (0.58, 2.89) | 1.32 (0.70, 2.48) | 1.30 (0.70, 2.41) | 0.70 (0.38, 1.31) |

CI, confidence interval; MHW, mental health and wellbeing; OR, odds ratio; PC, population centre.
[a]Odds Ratios from multivariable logistic regression model and adjusted for grade level, social and material deprivation quintiles.
[b]Defined as score ranging from −12 to 0 (inclusive), or factor scores <0.

from peers ceased. In addition, all schools were part of the APPLE Schools program that continued their health promotion activities to empower students in maintaining good health throughout the pandemic. While healthy eating activities such as taste testing and cooking classes were particularly challenging to deliver due to the COVID-19 safety protocols, a greater emphasis was placed on activities promoting mental health and wellbeing during this time [28, 29]. The APPLE Schools program had shared resources promoting mental health and wellbeing on schools' social media pages, offering online activities (e.g., guided meditations, virtual days focusing on positive messages), art-based assignments (e.g., window art, virtual music days and talent shows), support from mental health therapists, among others. Moreover, resilience at the community level may also have helped to mitigate the negative impact of the public health measures. In addition to common strategies (e.g., positive peer and family interactions, positive self-identity), children and youth living in rural and remote areas rely on community-based resilience that is grounded in community connectedness [30]. This fundamental feature of rural and remote communities has become prominent during the pandemic: in some cases, these communities were much more proactive in COVID-19 response compared to urban areas [31].

Strengths of this study are that we surveyed a specific segment of children residing in socioeconomically disadvantaged areas, and achieved high responses rates. Although the 12 items and their cumulative score demonstrated high internal consistency in previous studies [32], they were not designed to capture pandemic-specific stressors and the public health measures put in place. Instead, these items are suitable to administer in the general (rather than clinical) populations of school-aged children in school-based research, and serve as a precursor to early onset of mental health outcomes (depression, anxiety, conduct disorder, and attention-deficit/hyperactivity disorder) [33]. While we employed a repeated cross-sectional study design, which is the

design of choice when estimating trends over time, future research based on cohort studies of vulnerable children and youth will help reveal potential long-term negative psychosocial outcomes that might emerge from the stressors surrounding the pandemic, the limited social interactions and disrupted education [34]. Additionally, without a comparison or control schools, it is not possible to attribute the APPLE Schools health promotion programming to mental health and wellbeing outcomes. Lastly, we did not take into account seasonal effects that might have enhanced the comparability of two waves included in this study.

In sum, as children returned to school in Fall 2020 after the first COVID-19 lockdown was lifted, their mental health and wellbeing appear to return to pre-pandemic levels, similar to the levels observed among their peers pre-pandemic. The findings suggest that support for schools to tailor existing school-based health promotion programs and introduce new initiatives that encourage regular daily routine, healthy and active lifestyles, stimulate peer interaction and promote mental wellness in the school environment may help mitigate the effects of the pandemic on children's mental health and wellbeing. It is critical to ensure schools are equipped with adequate capacity (e.g., staff training, resources, funding) to deliver enhanced school programming and supports for student mental health and wellbeing throughout the pandemic. Similarly, supporting communities, particularly those located in disadvantaged areas, to reduce economic stressors [35] and to foster resilience may help minimize the impact of future pandemics or other disasters on children's mental health and wellbeing. These findings also align with recent calls that emphasize the importance of in-school learning for children's mental health and wellbeing, especially for older and disadvantaged children. While recognizing that school closures may be an important factor in controlling the COVID-19 outbreaks, the international organizations and professional pediatric societies urge that schools should remain open as long as possible during the pandemic response [36–40].

# ETHICS STATEMENT

The studies involving human participants were reviewed and approved by The Health Research Ethics Board of the University of Alberta (Pro00061528). Written informed consent for participation was not provided by the participants' legal guardians/next of kin because: All students provided assent and their parents/guardians provided active-information passive-permission consent. The Health Research Ethics Board of the University of Alberta (Pro00061528) and participating school boards approved all the procedures.

# AUTHOR CONTRIBUTIONS

All listed authors contributed to study design, drafted and revised the article, and gave their final approval of the version submitted for publication. KM and PV conceptualized the study and methodology, and secured funding and resources. MK, KM, and PV developed a statistical analysis plan, and MK conducted all data analyses. JD and PV accessed and verified the data and wrote the original draft. All authors reviewed and approved the final manuscript.

# FUNDING

The Public Health Agency of Canada (grant #1516-HQ-000071) along with other partners (for details see (41)) funded the APPLE Schools programming and data collection in the Northern communities. The present study was supported by operating funds from the Canadian Institutes for Health Research to KM and PV (grant #172685). KM holds the Murphy Family Foundation Chair in Early Life Interventions.

# CONFLICT OF INTEREST

The authors declare that the research was conducted in the absence of any commercial or financial relationships that could be construed as a potential conflict of interest.

# ACKNOWLEDGMENTS

The authors thank the students, parents/guardians, and school principals for their participation in the research. They further thank teachers, school health facilitators and champions, project assistants, the APPLE School staff for facilitating the research, and specifically Tina Skakun, Katherine Dekker and Landra Walker for their major roles in coordinating and conducting the data collection.

# SUPPLEMENTARY MATERIAL

The Supplementary Material for this article can be found online at: https://www.ssph-journal.org/articles/10.3389/ijph.2021.1604219/full#supplementary-material

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
