## [Reviewer comments · International Journal of Public Health]

Peer Review Report

Review Report on Mental health and wellbeing of elementary school children in Northern Canada before the COVID-19 pandemic and after the first lockdown

Original Article, Int J Public Health

Reviewer: Sally McManus

Submitted on: 16 May 2021

Article DOI: 10.3389/ijph.2021.1604219

EVALUATION

Q 1 Please provide your detailed review report to the authors. The editors prefer to receive your review structured in major and minor comments. Please consider in your review the methods (statistical methods valid and correctly applied (e.g. sample size, choice of test), is the study replicable based on the method description?), results, data interpretation and references. If there are any objective errors, or if the conclusions are not supported, you should detail your concerns.

There's a huge gap in research on whether COVID lockdown measures have had an impact on children's mental health and wellbeing. This study is therefore a welcome addition to a crucially important but under-researched area. It is very well written.

MAJOR

It would be helpful for the authors to include a line or two on the extent to which we can generalise from this cohort of young people. There are two conclusions made which aren't entirely compatible. Either our take-away from the paper should be that:

1. After the initial lockdown children appear to return to pre-pandemic wellbeing, or, that
2. The health promotion programme at these schools prevented the expected decline in mental health.

while I accept its impossible from the available data to tell which is the correct interpretation (and likely both do play a part) could the authors address this tension head on?

MINOR:

Title: for those of us not from Canada 'elementary school' may not be immediately meaningful. while the ages of the children are provided in the abstract, I couldn't see the ages given in the body of the paper (I may have missed this)? Likewise, what ages of child does 'grade 4', 'grade 5', 'grade 6' tend to equate to, perhaps average age rather than grade could be used in the table labels? This would help an international readership.

Sample: as well as the age range covered, is there information about ethnicity? How was the sample selected? were ALL the children in the relevant year groups eligible and therefore no selection was required, or were particular classes selected? Any information available on non-response - e.g. characteristics of children who took part compared with those who did not? This is not at all a deal breaker, such information may not be available. But if it is it would be useful to provide. Methods text needs to make clear that the comparisons are (rightly) based only on the 11 schools that took part in both waves. were weights applied to adjust for non-response?

Questionnaire: possible to add something about the rest of the questionnaire? The 12 mental health and wellbeing items are provided, were these the only questions asked or were these a part of a much longer questionnaire?

Avoid describing children as 'hard to reach' - there is some criticism of this phrase and the perspective it comes from.

Table 1/scoring: with the dichotomised score '-12 to 0' and '0 to 12' - labelling needs to clarify which group includes 0.

Figure: y axis needs labelling.

Limitations section needed acknowledging things which impact on comparability of the two survey waves, e.g. mode effects and seasonal effects. Seasonal effects are especially important in surveys of children as the timing of exams and holidays tend to occur at certain times of the year and are associated with wellbeing.

The authors may be interested in this national longitudinal survey of a probability sample of children and YP in England interviewed using the SDQ before and during lockdown:

[https://www.thelancet.com/journals/lanpsy/article/PIIS2215-0366\(20\)30570-8/fulltext](https://www.thelancet.com/journals/lanpsy/article/PIIS2215-0366(20)30570-8/fulltext)

However, I do agree with the authors that their study methods (repeated cross-sectional) are best for understanding population level change.

Q 2 Please summarize the main findings of the study.

It presents results from two cross-sectional surveys of children in the same group of schools in rural Canada, comparing spring 2018 and Nov-Dec 2020. The schools are all participating in a major health promotion programme. The focus on pre-pandemic compared with POST lockdown is useful, this framing has particular relevance as we look forward to whether there is long-term impact. The study finds little change (or very modest deterioration) in mental health, this is useful information. It's great that analyses are also run gender stratified. Where change was detected, this was largely confined to the oldest children.

Q 3 Please highlight the limitations and strengths.

As outlined above, the study does have some major limitations. The sample is very specific - regionally, socioeconomically, and because of an intervention - and its not quite clear therefore how we are to generalise from it. However, this is such an under researched area and data like this are so valuable, that it is important the study is published but just with limitations acknowledged.

Strengths include the timing, internal comparability of the pre and post sample, that the study is well-written, and also that it calmly and constructively presents 'negative' results (ie the lack of change in mental health). This is a valuable public health message.

PLEASE COMMENT

Q 4 Is the title appropriate, concise, attractive?

Title is fine except for the use of 'elementary school' which may not be meaningful to an international readership. Uses ages would be clearer.

Q 5 Are the keywords appropriate?

Yes.

Q 6 Is the English language of sufficient quality?

Yes, very well written.

Q 7 Is the quality of the figures and tables satisfactory?

Yes.

Q 8 Does the reference list cover the relevant literature adequately and in an unbiased manner?)

Yes.

QUALITY ASSESSMENT

Q 9 Originality

Q 10 Rigor

Q 11 Significance to the field

Q 12 Interest to a general audience

Q 13 Quality of the writing

Q 14 Overall scientific quality of the study

REVISION LEVEL

Q 15 Please take a decision based on your comments:

Minor revisions.